# IAG: Induction-Augmented Generation Framework for Answering Reasoning Questions

**Zhebin Zhang,**[*] **Xinyu Zhang,**[*] **Yuanhang Ren,**[*] **Saijiang Shi,**
**Meng Han, Yongkang Wu, Ruofei Lai, Zhao Cao**[†]
Huawei Poisson Lab, China
{zhangzhebin2, zhangxinyu35, renyuanhang}@huawei.com

## Abstract

Retrieval-Augmented Generation (RAG), by incorporating external knowledge with parametric memory of language models, has become the state-of-the-art architecture for open-domain QA tasks. However, common knowledge bases are inherently constrained by limited coverage and noisy information, making retrieval-based approaches inadequate to answer implicit reasoning questions. In this paper, we propose an **I**nduction-**A**ugmented **G**eneration (**IAG**) framework that utilizes inductive knowledge along with the retrieved documents for implicit reasoning. We leverage large language models (LLMs) for deriving such knowledge via a novel prompting method based on inductive reasoning patterns. On top of this, we implement two versions of IAG named IAG-GPT and IAG-*Student*, respectively. IAG-GPT directly utilizes the knowledge generated by GPT-3 for answer prediction, while IAG-*Student* gets rid of dependencies on GPT service at inference time by incorporating a student inductor model. The inductor is firstly trained via knowledge distillation and further optimized by back-propagating the generator feedback via differentiable beam scores. Experimental results show that IAG outperforms RAG baselines as well as ChatGPT on two Open-Domain QA tasks. Notably, our best models have **won the first place** in the official leaderboards of CSQA2.0 (since Nov 1, 2022) and StrategyQA (since Jan 8, 2023).

## 1 Introduction

Open-Domain Question Answering (ODQA) (Zhu et al., 2021) has attracted increasing research attention. Compared with the closed-domain setting, techniques developed upon ODQA models can empower search engines with the ability to respond to a wider range of user queries. As a typical knowledge-intensive task, ODQA has been extensively studied within the scope of information retrieval (Karpukhin et al., 2020; Das et al., 2019; Sachan et al., 2021), where access to external knowledge sources such as web pages or knowledge bases is required. Another line of research exploits large language models (LLMs) such as GPT-3 and PaLM as the knowledge source (Petroni et al., 2019; Liu et al., 2022b) and develops various prompting methods to elicit knowledge from LLMs that is implicitly encoded in the parameters. However, to answer reasoning questions, i.e., questions that demand some degree of reasoning ability, either retrieval-based or prompting-based approaches suffer from their intrinsic limitations.

On the one hand, although RAG (Lewis et al., 2020) has become the SOTA architecture for ODQA, documents retrieved from common knowledge bases generally suffer from the limitations of constrained coverage and noisy information, especially for implicit reasoning questions whose answers are not well documented. For example, it's trivial for an average child to answer questions such as *"can you catch a jellyfish in the dumpster?"*. However, it's unlikely to find the answer directly from the web or books. As shown in Figure 1, the retrieved documents can hardly answer the question in such cases. Hence, relying entirely on information retrieval is insufficient to solve implicit reasoning questions.

On the other hand, prompting-based methods can exploit the considerable amount of knowledge encoded in the parameters of LLMs for QA tasks (Wang et al., 2022; Wei et al., 2022; Chowdhery et al., 2022; Brown et al., 2020). But the problem of hallucination (i.e., generating natural language statements that are factually incorrect) imposes limitations on LLMs in terms of factuality and credibility. To better control the knowledge elicited from LLMs, various prompting methods such as chain-of-thought (CoT) (Wei et al., 2022) have been proposed by constructing intermediate reasoning steps

---

[*]These authors contributed equally to this work.
[†]Corresponding author.

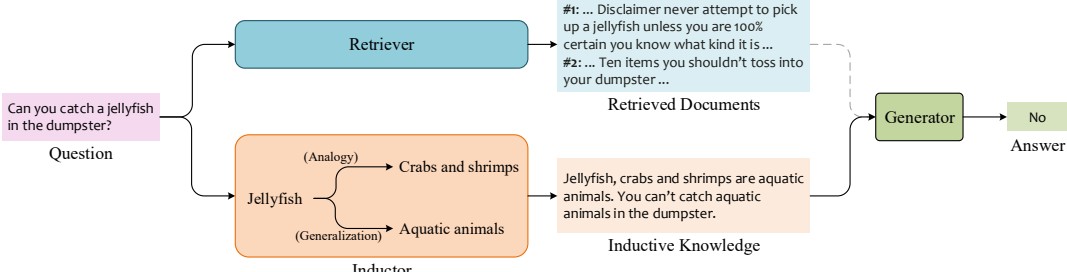

Figure 1: Overview of our IAG framework. The inductor provides inductive knowledge for predicting the answer when the retrieved documents are less informative.

until arriving at the conclusion. However, capability of the an LLM is intrinsically constrained by its parameter size, making it unable to respond correctly to domain-specific questions beyond the scope of its training corpus.

In view of the above challenges, it requires a new paradigm for building models applicable to reasoning QA. To this end, we combine the advantages of these two kinds of methods and propose **IAG**, an **I**nduction-**A**ugmented **G**eneration framework for answering implicit reasoning questions. IAG enhances the conventional RAG with an inductor that generates inductive knowledge w.r.t each question. To derive such knowledge, we propose a novel prompting method, which is intuitively inspired by the cognitive functions of inductive reasoning, to elicit inductive knowledge from an LLM (i.e., GPT-3). Our first implementation of IAG, which is termed IAG-GPT, directly leverages the knowledge statements sampled from GPT-3 as evidence alongside the retrieved documents to feed into the answer generator. We show that IAG-GPT improves over SOTA models on multiple ODQA datasets and has won the first place in the official leaderboards of CSQA2.0 and StrategyQA. We also implement IAG-*Student*, the second variant that gets rid of dependencies on GPT service during inference, by training a student inductor model following a two-step optimization scheme. Specifically, the model is firstly warmed up through distillation by using the GPT-3 statements as pseudo labels, and then further optimized with a novel TAILBACK approach: gradien**T** b**A**ck-propagation d**I**fferentiab**L**e **B**e**A**m s**C**ores feedbac**K**. TAILBACK implements a differentiable beam search algorithm for the inductor and allows the feedback from the generator to be back-propagated to the inductor via beam scores. We verify that IAG-*Student* improves over RAG baselines on a smaller architecture.

The contributions of this paper include: 1) an inductive prompting method which **improves the factuality of knowledge** elicited from LLMs by constructing a reasoning path via inductive reasoning; 2) an IAG-GPT implementation that **improves over strong baseline models and ChatGPT** by leveraging the knowledge elicited from GPT-3 as auxiliary supporting evidence for the generator; 3) a TAILBACK optimization algorithm to train the inductor which allows **IAG-*Student* to outperform RAG baselines under a small model size**.

## 2 Related Work

**Open-Domain Reasoning QA.** Retrieval-based approaches (Guu et al., 2020; Lewis et al., 2020) have been extensively explored to cope with ODQA tasks (Chen et al., 2017). Recent studies have focused on solving ODQA tasks that require certain sorts of reasoning, either explicitly or implicitly. Explicit reasoning tasks, such as Hotpot QA (bridge setting) (Yang et al., 2018), require the models to iteratively decompose the question and fetch new evidence until arriving at the answer. Implicit reasoning tasks, such as StrategyQA (Geva et al., 2021) and CSQA (without commonsense knowledge base) (Talmor et al., 2021), can be more difficult to solve since the answers are generally not present as plain text in web pages or knowledge bases. To solve implicit reasoning tasks, two kinds of methods have been proposed, i.e., RAG and prompting. The former retrieves evidence based on the decomposed questions and relies on the generator to implicitly reason over the evidence (Perez et al., 2020). But this approach becomes fragile when retrieved documents contain too little useful information or too much noise. The latter teaches LLMs to reason over questions by few-shot demonstration (Wei et al., 2022; Chowdhery et al., 2022). But prompting templates are generally task-specific

and LLMs are prone to hallucinations. Our proposed framework combines the advantages of these two methods.

**Prompting-Based Reasoning.** Few-shot prompting is a mainstream approach to eliciting knowledge from LLMs (Brown et al., 2020; Chowdhery et al., 2022; Ma et al., 2022; Liu et al., 2022a). To answer implicit reasoning questions, various prompting methods have been proposed among which CoT (Wei et al., 2022) has attracted the most research interest in the earlier research. It allows LLMs to answer complex questions in a step-by-step manner. Follow-up studies include enhancing the consistency of CoT (Wang et al., 2022) and improving the correctness of the reasoning steps. Recently, adaptive prompting methods have been shown to achieve better performance on reasoning tasks such as bABI (Weston et al., 2016) and ProofWriter (Tafjord et al., 2021). For example, the Selection-Inference (SI) framework (Creswell et al., 2022) alternates between selection and inference to generate a series of interpretable reasoning steps. LAMBADA (Kazemi et al., 2022) proposes the idea of backward chaining and employs four LLM-based sub-modules for reasoning. However, SI and LAMBDA rely on heuristic search algorithms performed in a limited context space, which cannot be easily adapted to open-domain reasoning tasks. The inductive prompting proposed in this paper can be used for general-purpose knowledge elicitation.

## 3 Induction-Augmented Generation

### 3.1 Overview

As illustrated in Figure 1, IAG enhances the conventional RAG architecture with an inductor that provides inductive knowledge for the generator to predict the answer. The inductor takes the question as input and outputs knowledge statements in the form of inductive reasoning. These statements, together with the retrieved documents, are used as supporting evidence to feed into the generator.

Section 3.2 introduces the inductive prompting method used for enhancing the factuality of the knowledge elicited from LLMs. Two implementations of our proposed IAG framework, i.e., IAG-GPT and IAG-*Student* are present in Section 3.3.

### 3.2 Knowledge Elicitation via Inductive Prompting

Recently, exploiting LLMs (e.g., GPT-3) for QA tasks has attracted increasing research interest due to their abilities to store factual knowledge and perform reasoning. Representing knowledge as free-form text qualifies LLMs as high-coverage knowledge bases for various domains, but the knowledge elicited from LLMs is prone to factual errors that can be detrimental to downstream tasks. Existing approaches to this problem either focus on arithmetic reasoning or commonsense reasoning tasks that involve multiple reasoning steps (Wei et al., 2022) or cannot be easily adapted to open-domain settings (Creswell et al., 2022; Kazemi et al., 2022).

To enhance the credibility of the statements generated by LLMs, we propose a prompting method that is intuitively inspired by the idea of inductive reasoning. Inductive reasoning is a method of logical thinking that draws general conclusions from specific observations, during which **analogy** and **generalization** are two fundamental cognitive tools. Analogical reasoning allows us to establish connections between two similar objects and infer new information about one object based on what is known about the other. Generalization involves exploiting category information to generalize from the known to the unknown. As an example, consider the question shown in Figure 1. By analogical reasoning, one might conjecture that jellyfish, just like crabs and shrimps, are seldom found in dumpsters because these animals are similar in terms of inhabitants. By generalization, the fact that jellyfish are aquatic animals can support the hypothesis that they live in the water instead of in dumpsters.

Based on the above cognitive functions, we propose a prompting method that guides LLMs to generate knowledge statements w.r.t. the question by building a two-step reasoning path, formally given as:

**Question:** A statement about *target*.
**Knowledge:** *Target*, *analog#1*, *analog#2* are *hypernym*. An assertion about *hypernym*.

The knowledge statement is formulated as a reasoning path consisting of two sentences. The first sentence categorizes the *target* into its conceptual *hypernym* by associating two *analogs* that share some commonalities. And the second sentence states a fact about the *hypernym* that is contextually related to the topic of the question.

As an example, consider the question shown in Figure 1. Regarding jellyfish as *target*, its *analogs* are crabs and shrimps, and its *hypernym* is aquatic animals. Hence, the reasoning path can be constructed as follows: *Jellyfish, crabs and shrimps are aquatic animals. You can't catch aquatic animals in the dumpster.* We follow the above inductive reasoning pattern and construct a prompting template consisting of 5 in-context demonstrations. The template is presented in Appendix A.

### 3.3 IAG Implementations

#### 3.3.1 IAG-GPT

For IAG-GPT, the function of induction is fully delegated to the GPT-3 service API. We leverage the prompting method described in Section 3.2 to generate inductive knowledge for each question. Inspired by (Wang et al., 2022), we proposed to enhance the validity of the result by aggregating multiple knowledge statements. However, instead of explicitly voting on multiple results, we let the generator implicitly reason over the collection of all sampled knowledge statements.

Specifically, for each question, we sample $M$ inductive knowledge statements from GPT-3 and append them to the $N$ retrieved documents, leading to a collection of $M + N$ pieces of evidence. The generator, during both training and inference, takes all the $M + N$ evidence as input following the fusion-in-decoder (FiD) approach (Izacard and Grave, 2021).

There are two benefits to feeding multiple sampled knowledge statements into the generator. During training, the generator learns to predict the correct answer based on a diverse set of knowledge statements, making it robust to the noise present in the provided evidence. During inference, providing a richer set of evidence avoids the problem of local optimality in greedy decoding and has a better chance of yielding the knowledge statement that best prompts the generator.

#### 3.3.2 IAG-*Student*

To get rid of the dependencies on GPT-3 during inference, we replace GPT-3 with a student inductor model (we refer to it as inductor for brevity when there's no confusion). The inductor is trained in two steps. In the first step, we warm it up via a distillation method, i.e., the knowledge statements elicited from GPT-3 are used as pseudo labels to train the inductor with a seq2seq loss. In the second step, we perform further optimization using

a novel scheme TAILBACK that back-propagates the feedback from the generator to the inductor via differentiable beam scores.

**Step 1: Distillation** For each question-answer pair $(q, a^*)$ in the training set, we sample $N$ different knowledge statements from GPT-3 using inductive prompting described in Section 3.2. The generated statements for the question are denoted as $\mathcal{K} = \{K_n\}_{n=1}^N$. Besides, each question-answer pair is accompanied by the top-$M$ documents ranked by the retriever, represented as $\mathcal{R} = \{R_m\}_{m=1}^M$.

Instead of directly supervising the inductor using all the knowledge statements generated by GPT-3, we claim that different statements should be distinguished according to their respective confidence during distillation. The confidence of each statement can be measured by the probability of predicting the ground-truth answer when used as supporting evidence for the generator.

Firstly, we adapt the generator to the task-specific dataset by fine-tuning for a few steps using only the retrieved documents as supporting evidence. To calculate the confidence values of the $N$ knowledge statements, we fed each of them as extra supporting evidence alongside $M$ retrieved documents into the generator, and compute the probability of the ground-truth answer as

$$p_n = p(a^*|\theta_{gen}, q, K_n, \mathcal{R}), n \in [1, N], \quad (1)$$

where $\theta_{gen}$ is the parameters of the generator. We derive the confidence values $\{\tilde{p}_n\}_{n=1}^N$ by normalizing the above probability distribution $\{p_n\}_{n=1}^N$ following

$$p'_n = \frac{p_n - \mu}{\sigma}, \quad \tilde{p}_n = \frac{\exp(p'_n)}{\sum_{i=1}^N \exp(p'_j)}, \quad (2)$$

where $\mu$ and $\sigma$ are the mean value and standard deviation of the distribution $\{p_n\}_{n=1}^N$, respectively. We found that the normalized values better reflect the difference among the knowledge statements.

Finally, we train the inductor using the knowledge statements from GPT-3, taking into account their confidence values. We propose two distillation strategies that construct different distillation loss. The first strategy, termed $\mathcal{Q}_{\text{Max}}$, allows only one knowledge statement for each question (i.e., the one with the maximum confidence) to contribute to the loss, which is formulated as

$$\mathcal{L}_{Max} = - \sum_{y_t \in K_{\hat{n}}} \log(p(y_t|\theta_{ind}, q, y_{<t})), \quad (3)$$

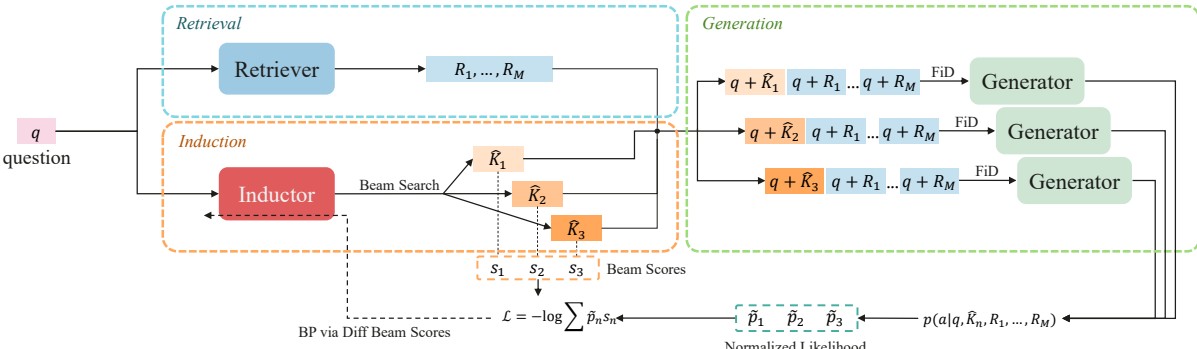

Figure 2: Illustration of our proposed TAILBACK training scheme. The likelihood values of the ground-truth answer conditioned on different knowledge statements are used as feedback from the generator, which can be back-propagated to the inductor via differentiable beam scores.

where $\hat{n} = \text{argmax}_n \tilde{p}_n$ and $\theta_{ind}$ represents the parameters of the student inductor model.

For the second strategy $\mathcal{Q}_{\text{Weight}}$, all the $N$ statements of each question can contribute to the total loss, but are weighted by their confidence values, given by

$$\mathcal{L}_{\text{Weight}} = -\sum_{n=1}^{N} \tilde{p}_n \left( \sum_{y_t \in K_n} \log(p(y_t|\theta_{ind}, q, y_{<t})) \right). \quad (4)$$

**Step 2: TAILBACK**  After warmed up through distillation, the inductor is further optimized using a novel TAILBACK training scheme that back-propagates the end-to-end loss to the inductor via differentiable beam scores, as depicted in Figure 2.

Intuitively, TAILBACK calculates the prediction loss of the generator based on the knowledge statements produced by the inductor and steers the model toward generating the statement that yields minimal prediction loss. A major challenge of this training scheme is that the generator is conditioned on a discrete sequence of tokens produced by the student inductor model via auto-regressive decoding, which prevents the end-to-end loss from updating the parameters of the inductor through gradient descending.

To address this issue, we implement a **differentiable beam search** algorithm for the inductor where the beam scores can be back-propagated. Specifically, the differentiable algorithm differs from the default implementation in that it preserves the computational graphs for deriving the beam scores instead of converting them into non-differentiable scalars during computing. Hence, the gradients are allowed to be propagated through the inductor via the beam scores.

**Algorithm 1** TAILBACK Optimization

**Input:**
  $\mathcal{D}$: training dataset
  $T$: number of training iterations
1: **for** $t \leftarrow 1$ to $T$ **do**
2:   $(q, a*) \leftarrow get\_sample(\mathcal{D})$
3:   $\{\hat{K}_n, s_n\}_{n=1}^{N} \leftarrow beam\_search(\theta_{ind}, q)$
4:   $\{R_m\}_{m=1}^{M} \leftarrow get\_document(q)$
5:   **for** $n \leftarrow 1$ to $N$ **do**
6:     $p_n \leftarrow p(a * |\theta_{gen}, q, K_n, \{R_m\}_{m=1}^{M})$
7:   **end for**
8:   $\{\tilde{p}_n\}_{n=1}^{N} \leftarrow normalize(\{p_n\}_{n=1}^{N})$
9:   $loss \leftarrow \sum_{i=1}^{N} \tilde{p}_i \cdot s_i$
10:   $loss.backward()$
11: **end for**

Now we can describe the workflow or the TAILBACK optimization scheme. The pseudo-code can be found in Algorithm 1. First, given a question $q$, we use differentiable beam search to generate $N$ knowledge statements denoted as $\hat{\mathcal{K}} = \{\hat{K}_n\}_{n=1}^{N}$. We normalize the beam scores of $N$ knowledge statements using softmax, denoted as

$$s_n = \text{softmax}(p(\hat{K}_n|\theta_{ind}, q)), n \in [1, N]. \quad (5)$$

Then, we derive the end-to-end feedback from the generator by computing the probability of predicting the ground-truth answer given the question, $M$ retrieved documents, and each generated knowledge statement. Finally, the loss of the student inductor model is constructed as

$$\mathcal{L}_{\text{TAILBACK}} = -\log \sum_{n=1}^{N} \mathbb{SG}[p(a^*|\theta_{gen}, \\ q, \hat{K}_n, \mathcal{R})] \cdot p(\hat{K}_n|\theta_{ind}, q). \quad (6)$$

where $\mathbb{SG}[.]$ is the stop gradient operator, i.e. the parameters of the generator won't be updated during the optimization of the inductor.

After TAILBACK optimization, we sample $N$ different knowledge statements from the inductor and use them together with the retrieved documents to fine-tune the generator, just the same way as IAG-GPT.

## 4 Experimental Setup

### 4.1 Datasets

We evaluate IAG on two open-domain QA benchmarks, namely, CSQA2.0 (Talmor et al., 2021) and StrategyQA (Geva et al., 2021), both of which are binary classification tasks. **CSQA2.0** consists of 14343 examples about everyday commonsense knowledge (9264/2541/2473 for train/dev/test), and **StrategyQA** is a multi-hop QA task comprised of 2780 examples (2290/490 for train/test) which requires implicit reasoning to solve. Note that since the official **StrategyQA** dataset doesn't include a development set, we randomly draw 1/4 of the examples out of the training set for evaluation.

### 4.2 Models

This section describes our setups for different components of IAG.

**Retriever.** For StrategyQA, the retriever is implemented as a sparse BM25 algorithm followed by a single-stream reranker (Gao et al., 2021). The dataset is accompanied by a corpus of context documents as well as several annotated facts corresponding to each question, which are used to optimize the reranker model. For CSQA2.0, we input the questions into Google Search and use the top-5 snippets of the returned results as the retrieval data.

**Inductor.** For IAG-GPT, the inductor is implemented by invoking the GPT-3 service API (text-davinci-003). We employ a sampling method with a temperature of 0.7. For IAG-*Student*, we adopt T5-Large as the backbone of the student inductor model.

**Generator.** For IAG-GPT, we initialize the generator with a T5-11B backbone. For IAG-*Student*, since fitting the inductor and the generator modules pose high requirements on the GPU memory beyond our hardware capabilities, we resort to a smaller architecture and use T5-Large for the generator as well.

## 5 Results

### 5.1 Main Results

We first report the performance of IAG-GPT in comparison with SOTA models, as presented in Table 1. It shows that **the improvement of IAG over SOTA (74.1 → 78.2 for CSQA2.0 and 69.4 → 72.9 for StrategyQA) is significant compared with the improvement of SOTA over previous SOTA (72.4 → 74.1 for CSQA2.0 and 65.4 → 69.4 for StrategyQA).**

For IAG-GPT, we report the scores of three different setups of supporting evidence fed to the generator. IAG-GPT outperforms existing methods on CSQA2.0 and StrategyQA. Besides, scores on the randomly held-out subsets of two tasks show that IAG-GPT has **significant advantages over ChatGPT** (version up to Jan 29, 2023) in answering reasoning questions. This result suggests that LLMs, even as strong as ChatGPT, can suffer from hallucination problems without access to retrieved factual evidence. Hence, combining prompting methods with information retrieval empowers IAG-GPT to answer reasoning questions more reliably. We show in Table 8 some cases where inductive knowledge helps predict the answer.

Notably, our most powerful models have won the first place on the official leaderboards of CSQA2.0 and StrategyQA. Specifically, our IAG-GPT using 10 retrieved documents and 5 knowledge statements, enhanced with an assembling method, has set a new record of **78.08**[*] on CSQA2.0. For StrategyQA, the single model of IAG-GPT using 5 retrieved documents and 5 knowledge statements achieves a SOTA score of **72.86**[†].

### 5.2 Prompting Methods

To verify the effectiveness of inductive prompting, we compare IAG-GPT with two other baseline prompting methods. The first baseline is an ablated version of IAG-GPT that elicits knowledge from GPT-3 using a trivial prompt without inductive reasoning. The template for trivial prompting is presented in Table 6. The second baseline directly derives answers from GPT-3 via chain-of-thought (CoT) prompting using the prompt proposed in their original paper (Wei et al., 2022). For the second baseline, we experiment with two implemen-

---

[*]https://leaderboard.allenai.org/csqa2/submissions/public
[†]https://leaderboard.allenai.org/strategyqa/submissions/public

Table 1: Performance on two ODQA tasks. The first two columns report scores on CSQA2.0 dev set and StrategyQA test set respectively. The last two columns compare IAG with ChatGPT on a randomly held-out subset containing 50 examples for each task.

| Method | | CSQA2.0 dev | StrategyQA test | CSQA2.0 dev* | StrategyQA dev* |
|---|---|---|---|---|---|
| DisentangledQA (Liu et al., 2022c) | | - | 69.4 | - | - |
| UL2 (Tay et al., 2022) | | - | 59.0 | - | - |
| Auto-CoT (Zhang et al., 2022) | | - | 65.4 | - | - |
| UNICORN-11B (Talmor et al., 2021) | | 74.0 | - | - | - |
| T5-11B (Talmor et al., 2021) | | 74.1 | - | - | - |
| GKP (Liu et al., 2022b) | | 72.4 | - | - | - |
| ChatGPT (Ouyang et al., 2022) | | - | - | 60.0 | 52.0 |
| IAG-GPT (T5-11B Generator) | Retrieval Only | 77.2 | 70.0 | - | - |
| | Induction Only | 74.0 | 71.2 | - | - |
| | Retrieval & Induction | **78.2** | **72.9** | **80.0** | **74.0** |

tations including 1) the original implementation based on greedy decoding and 2) an improved variant (Wang et al., 2022) that aggregates 5 reasoning traces via majority voting.

Table 2: Comparison between IAG-GPT and CoT prompting. Scores are reported for the StrategyQA dev set.

| Method | | Accuracy |
|---|---|---|
| CoT | Greedy | 71.5 |
| | Self-Consistency | 73.3 |
| IAG (Trivial) | w/o retrieval | 73.6 |
| | w/ retrieval | 74.8 |
| IAG (Inductive) | w/o retrieval | 75.5 |
| | w/ retrieval | **76.2** |

The experiment is conducted on the StrategyQA dev set and the scores are reported in Table 2. As shown, our proposed IAG framework outperforms CoT methods by a large margin. This result indicates that compared to relying entirely on GPT-3 for answer prediction, combining the knowledge elicited from GPT-3 with information retrieval is a better way of utilizing LLMs for QA tasks. Also, we find that inductive reasoning is better than trivial prompting at providing the generator with useful knowledge for answer prediction. Table 9 lists some cases comparing different methods.

**Limitations of Inductive Prompting.** Although inductive prompting proves to be more effective in producing insightful knowledge, we have to admit

that it can fail and generate faulty statements sometimes. Take the third question listed in Table 8 as an example, GPT-3 claims that *"Beverages are not capable of drowning someone"*. As another example, given the question "Cotton candy is sometimes made out of cotton?", GPT-3 generates the following statements *"Cotton, wool, and silk are fabrics. Cotton candy is made out of spun sugar or fabric"*. We attribute these failures to the fact that, although indutive prompting helps establish the connection between a concept and its hypernym, correctly predicting a fact related to the hypernym still depends on the internal knowledge of the language model, which is error-prone for tricky or long-tail questions.

### 5.3 Optimization of Inductor

For IAG-*Student*, the inductor model is optimized following the two-step training scheme as described in Section 3.3.2. This experiment provides insights into this training process.

#### 5.3.1 Distillation Strategies

In Section 3.3.2, we propose two strategies for the warmup training of the inductor, i.e., $\mathcal{Q}_{\text{Max}}$ and $\mathcal{Q}_{\text{Weight}}$. A prior study (Liu et al., 2022a) employs a straightforward strategy that uses all these statements for training, denoted as $\mathcal{Q}_{\text{All}}$. The end-to-end performance of IAG-*Student* is evaluated by adopting three different distillation strategies. We also introduce an RAG baseline that doesn't leverage inductive knowledge for prediction.

As shown in Figure 3, $\mathcal{Q}_{\text{All}}$ hardly outperforms the RAG baseline ($64.3 \rightarrow 64.5$). This can be at-

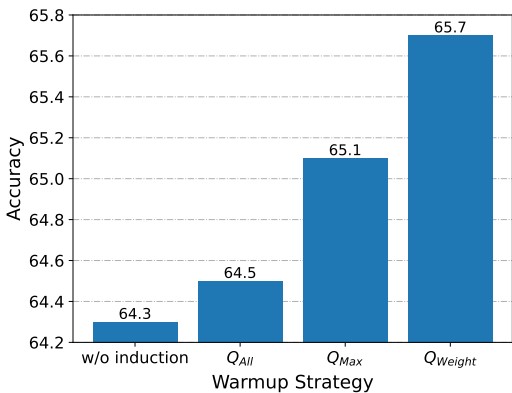

Figure 3: Comparison of different warmup strategies. Scores are reported on the StrategyQA dev set.

tributed to the fact that some knowledge statements sampled from GPT-3 are less useful for answer prediction. Using these statements indiscriminately can inevitably introduce noise and disturb the optimization of the inductor. As for $\mathcal{Q}_{\mathrm{Max}}$, leveraging feedback from the generator allows the inductor to learn from the best knowledge statement. But the size of the training data set is much smaller (only a fraction of $\frac{1}{N}$ generated statements are used). In comparison, $\mathcal{Q}_{\mathrm{Weight}}$ achieves the best performance among all the strategies by supervising the inductor with a more diverse set of GPT-3 statements while suppressing the effect of low-contribution ones. Hence $\mathcal{Q}_{\mathrm{Weight}}$ is adopted for the rest of the experiments.

### 5.3.2 TAILBACK

To provide insightful analysis of the TAILBACK optimization scheme, we evaluate the performance of IAG-*Student* at three different stages: 1) without any induction knowledge, 2) introducing the student inductor model trained by distillation, and 3) further optimizing the model with TAILBACK. Note that both the inductor and the generator adopt the T5-Large architecture in this experiment.

Table 3: Performance of IAG-*Student* at different stages on the dev sets of CSQA2.0 and StrategyQA.

| Training Step | CSQA2.0 | StrategyQA |
|---|---|---|
| Retrieval Only | 61.8 | 64.3 |
| + Distillation | 60.5 (-1.3) | 65.7 (+1.4) |
| + TAILBACK | **61.9 (+0.1)** | **66.6 (+2.6)** |

It can be shown in Table 3 that, introducing the inductor model significantly promotes the performance on StrategyQA, whereas little improvement

is observed on CSQA2.0. Since most questions in CSQA2.0 can be readily answered by using the retrieved documents, the inductive knowledge become less useful. However, solving StrategyQA requires implicit reasoning over multiple documents. When the retrieved documents fail to provide useful clues, our inductor can compensate for the missing information.

Nevertheless, we observe consistent performance improvements brought by the further TAILBACK training on top of distillation on both CSQA2.0 (60.5 → 61.9) and StrategyQA (65.7 → 66.6). This result proves that TAILBACK indeed steers the inductor towards producing knowledge statements that better prompt the generator. Qualitative analysis is provided in Table 10.

### 5.4 Knowledge Fusion Mechanism

#### 5.4.1 Knowledge Fusion v.s. Self-Consistency

IAG fuses multiple sampled knowledge statements into the generator for prediction. In contrast, the self-consistency approach (Wang et al., 2022) proposes to explicitly vote on different reasoning paths sampled from LLM. We implement the idea of self-consistency on IAG by allowing the generator to predict multiple answers based on individual knowledge statements and reach a consensus by majority voting.

Table 4: Comparison between knowledge fusion and self-consistency.

| Method | CSQA2.0 dev | StrategyQA dev |
|---|---|---|
| w/o induction | 77.2 | 72.7 |
| Self-Consistency | 77.6 | 74.8 |
| Knowledge Fusion | **78.2** | **76.2** |

Table 4 compares different methods including the retrieval-only baseline. The knowledge fusion mechanism is obviously superior to the self-consistency approach. We hypothesize that fusing all knowledge statements allows the generator to have a holistic view of all evidence and make informed decisions. As for self-consistency, although the voting mechanism can eliminate minority errors to some extent, it's less reliable due to easier propagation of random errors in the sampled statements to the conclusions.

### 5.4.2 Number of Knowledge Statements

Our implementation of IAG samples 5 knowledge statements to feed into the generator. To justify this design choice, we evaluate the performance of IAG-*Student* with varying statement numbers. As shown in Figure 4, IAG-*Student* achieves the best performance with the statement number between 5 and 7. The performance drops when the sampling number is either too large or too small. On the one side, a small sampling number makes the model prone to random errors. On the other side, sampling too many statements inevitably introduce more noisy information that could mislead the prediction. Hence our choice of 5 knowledge statements makes a reasonable trade-off.

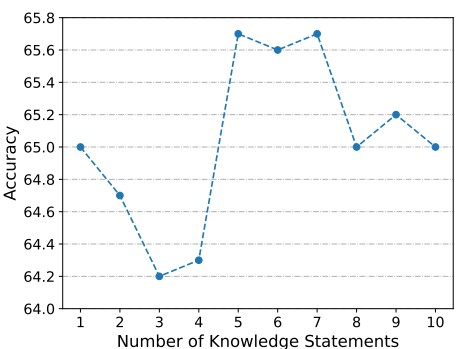

Figure 4: Scores of IAG-*Student* on StrategyQA dev set with different numbers of knowledge statements.

## 6  Conclusion

To tackle the problem that retrieval-based methods cannot provide sufficient knowledge for the generator to answer implicit reasoning questions, we propose a novel IAG framework that augments RAG with inductive knowledge elicited from language models. We first design an inductive prompting method that enhances the factuality of knowledge elicited from GPT-3. We further propose a sophisticated optimization scheme that trains a student inductor model via distillation and TAILBACK. Our results suggest that IAG outperforms RAG in answering implicit reasoning questions.

## Limitations

This work has several limitations. First, IAG has evident advantages over RAG only for questions that cannot be readily answered by the retrieved documents. Otherwise, the performance boosts brought by providing inductive knowledge are less significant. Second, the effectiveness of IAG-*Student*

and the proposed distillation and TAILBACK optimization scheme has only been verified on the T5-Large architecture. We leave it as our future work to experiment our methods with larger models and various backbones.

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

## A  Prompting Template

Our experiments used two prompting templates including an inductive The template used for inductive prompting is presented in Table 5. It consists of 5 demonstrations constructed based on inductive reasoning, appended by the question of interest. We also present in Table 6 the trivial prompting template that is used in Section 5.2.

## B  Additional Experimental Results

### B.1  Comparison between Information Retrieval and knowledge Induction

Table 8 lists some cases where the retrieved documents fail to provide informative evidence whereas the inductive knowledge stands out. Typical failures of retrieval-based approaches can be categorized into two occasions. Firstly, when the knowledge required for answering the question is too scarce to be found, the retriever could fetch documents that hardly match the semantics of the question (e.g., Q3). Secondly, when the question is too trivial and the answer is unlikely to be officially documented, the retrievals could contain specific cases that contradict the commonsense conclusion (e.g., *brittle failure of steel* for Q1 and *unequal leg length* for Q2).

Besides, the results on different setups of IAG-GPT suggest that, the relative contributions of the retrieval and the inductive knowledge can be different, depending on the tasks. As shown in Table 1, for CSQA2.0, higher scores are reported for retrieval only than for induction only, while the result is contrary for StrategyQA. These results can be attributed to the fact that questions in StrategyQA (e.g., *Would a cattle farmer be useful to a drum maker?*) require better reasoning ability to answer.

To fully verify the effectiveness of the inductive knowledge, we compare the performance of IAG based on two different settings: 1) using 10 retrievals and 2) using 5 retrievals and 5 knowledge statements. The results are shown in Table 7. For CSQA2.0, using more retrievals yields a slightly better result, while for StrategyQA, introducing knowledge statements significantly boosts the performance. This finding is consistent with our previous conclusion that our method works better for reasoning-intensive QA tasks. For CSQA2.0, many commonsense questions can be solved by referring to plain-text records by retrieving methods. However, for StrategyQA, the answers are unlikely to be directly extracted from the retrieved documents but require reasoning over multiple documents to obtain. In such cases, introducing inductive knowledge can be very useful.

This is also the reason why the marginal performance gains are more significant for StrategyQA than for CSQA2.0 when inductive knowledge is offered besides the retrieved documents.

### B.2  Comparison among Prompting Methods

Table 9 compares the reasoning traces of different prompting methods. Although CoT is regarded a reliable prompting method by constructing intermediate reasoning steps, it's also prone to errors that could occur at any point of the reasoning path. For example, the sentence *Thus, Christmas trees are not dissimilar to deciduous trees* is a false deductive result based on the previous statements, leading to a false conclusion. As shown in Table 2, although enhancing CoT with self-consistency improves performance ($71.5 \rightarrow 73.3$), the intrinsic problem identified above still plagues this approach. Let alone the advantage that IAG can utilize retrieved documents as extra information for prediction (76.2), the ablated version that employs only the GPT-3 knowledge (75.6) still has a lead over CoT.

Table 5: The inductive prompting template used in the experiments.

---

Question: It is safe to keep wolves as pets.
Knowledge: Wolves, tigers and lions are wild animals. Wild animals are generally dangerous.
###
Question: Bacon is healthy diet food.
Knowledge: Bacon, chips and cakes are junk food. Junk food is not healthy.
###
Question: Pens are more expensive than cars.
Knowledge: Pens, erasers and paper are stationery. Stationery is cheaper than cars.
###
Question: People make furniture out of oak.
Knowledge: Oak, pine and beech are Wood. Wood can be used to make furniture.
###
Question: Fridges are often used in the wild.
Knowledge: Fridges, ovens and TVs are appliances. Appliances are used in houses.
###
Question: {User Question}
Knowledge:

---

Table 6: The trivial prompting template used in the experiments.

---

Question: It is safe to keep wolves as pets.
Knowledge: Wolves are dangerous.
###
Question: Bacon is healthy diet food.
Knowledge: Bacon is not healthy.
###
Question: Pens are more expensive than cars.
Knowledge: Pens are cheaper than cars.
###
Question: People make furniture out of oak.
Knowledge: Oak can be used to make furniture.
###
Question: Fridges are often used in the wild.
Knowledge: Fridges are used in houses.
###
Question: {User Question}
Knowledge:

---

Table 7: Performance comparison between different settings on CSQA2.0 and StrategyQA dev sets.

| Setting | CSQA2.0 | StrategyQA |
|---|---|---|
| 10 retrievals | 78.4 | 73.8 |
| 5 retrievals + 5 knowledge | 78.2 | 76.2 |

still suffers from factual errors (e.g., #D3 for Q1), which is probably due to the limitation imposed by a small model size. In comparison, further optimization via TAILBACK deviates the inductor from the pre-defined reasoning pattern, but the knowledge (e.g., #T1 for Q2) can better guide the generator in predicting the right answer.

## B.3 Effect of Inductor Optimization

We list some cases in Table 10 that compares the knowledge statements generated by the student inductor model at different optimization stage. As shown, after TAILBACK training, the inductor produces knowledge statements that are more consistent. Besides, we find that, although distillation training enables the inductor to grasp the inductive reasoning pattern, the generated knowledge

Table 8: A demonstration of knowledge elicited from GPT-3 for cases in the CSQA2.0 dev set. Text colored in red indicates factual errors or contradictions to the ground-truth answer, and green indicates supporting evidence.

| Q & A | Retrieved Documents | Inductive Knowledge |
|---|---|---|
| **Q1**:A tube is never brittle if it is made of steel. 
 **Answer**: Yes 
 **Prediction (w/o induction)**: No 
 **Prediction (w/ induction)**: Yes | **#1**. Under what circumstances does **the brittle failure of steel** take place? 
 **#2**. **The material becomes brittle** and, in extreme cases, mere contact with molten metal ... | **#1**. Steel is a material. Materials can vary in their properties, including brittleness. 
 **#2**. Steel, iron and concrete are metals. **Metals are not brittle.** |
| **Q2**:A person has legs the same size as each other. 
 **Answer**: Yes 
 **Prediction (w/o induction)**: No 
 **Prediction (w/ induction)**: Yes | **#1**. **Unequal leg length** is where the legs are either different lengths or ... 

 **#2**. **Having one leg longer than the other is moderately normal.** The condition is ... | **#1**. Legs, arms and fingers are appendages. All appendages on a person **are usually the same size.** 
 **#2**. People have two legs. **Legs are the same size** as each other. |
| **Q3**:Juice is capable of drowning someone. 
 **Answer**: Yes 
 **Prediction (w/o induction)**: No 
 **Prediction (w/ induction)**: Yes | **#1**. A physical dissertation on drowning: in which submersion, commonly ... 
 **#2**. Because infants and young children can drown in only a few inches of water, even water-filled containers ... | **#1**. Juice, water and soda are liquids. **Liquids can drown people.** 
 **#2**. Juice, water and milk are beverages. **Beverages are not capable of drowning someone.** |

Table 9: Cases from StrategyQA to demonstrate the differences among the three prompting methods. Green indicates statements that support the answer, while red indicates otherwise.

| Q & A | CoT Prompting | Trivial Prompting | Inductive Prompting |
|---|---|---|---|
| **Q1**: Are Christmas trees dissimilar to deciduous trees? 
 **Answer**: Yes | Christmas trees are **evergreen trees**. Deciduous trees are **tree that lose their leave**. Thus, Christmas trees are **not dissimilar** to deciduous trees. So the answer is no. | Christmas trees are **conifer trees**. Deciduous trees are **trees that lose their leaves** each year. | Christmas trees, fir trees and pine trees **are coniferous trees. Deciduous trees are not coniferous.** |
| **Q2**: Do frogs feel disgust? 
 **Answer**: Yes | Disgust is a human emotion. There is **no evidence that frogs feel emotions**. Thus, frogs don't feel disgust. | Scientists **do not currently know whether frogs feel emotions.** | Frogs, lizards and snakes are animals. **Animals can feel emotions such as disgust.** |
| **Q3**: Is the foot part of the metric system? 
 **Answer**: No | The metric system uses meters. 1 meter is about 3.3 feet. **Thus, the foot is part of the metric system.** So the answer is yes. | The foot is **not part of the metric system.** | The foot, inch and yard **are Imperial units. Imperial units are not part of the metric system.** |

Table 10: Comparison of inductive knowledge generated by the student inductor at different optimization stages. Text colored in red indicates factual errors or contradictions to the ground-truth answer, and green indicates supporting evidence.

| Q & A | Inductive Knowledge after Distillation | Inductive Knowledge after TAILBACK |
|---|---|---|
| **Q1**: Does ontology require a scalpel? **Answer**: No | **#D1**. Ontology is the study of the human mind. It requires knowledge and understanding. **A scalpel is not required for ontology.** **#D2**. Ontology is the study of the inner workings of human beings. **It requires a scalpel.** **#D3**. Ontology, astronomy and philosophy are science disciplines. **Science disciplines require a scalpel.** | **#T1**. Ontology is the study of philosophy and religion. **Philosophy and religion don't require a scalpel.** **#T2**. Ontology is a branch of knowledge that studies the meaning and structure of human thought and behavior. **It does not require a scalpel to be used.** **#T3**. Ontology is a branch of science that deals with knowledge and understanding. **Knowledge is not usually obtained with a scalpel.** |
| **Q2**: Can a person be diagnosed with a Great Depression? **Answer**: No | **#D1**. **A Great Depression is a depression.** Depressions do not affect everyone. **#D2**. The Great Depression is a period of extreme economic uncertainty. **People with extreme financial difficulties can be diagnosed with a Great Depression.** **#D3**. The Great Depression was a time of great economic change in the 1930s. **A person can be diagnosed with a Great Depression.** | **#T1**. People often think of the Great Depression as **an economic crisis, not a mental illness.** **#T2**. A Great Depression is a period of economic downturn in the United States. Economic downturns often happen during the Great Depression. **#T3**. Great Depression, post World War II Depression and postwar economic depression are all periods of economic downturn. Economic downturns can have drastic effects on individuals and their families. |