# OpenReview forum: "IAG: Induction-Augmented Generation Framework for Answering Reasoning Questions"
_EMNLP/2023/Conference — EMNLP 2023 Main_

### Official Review · Reviewer_HrRJ · 2023-08-03

**Soundness:** 4

**Excitement:**

4: Strong: This paper deepens the understanding of some phenomenon or lowers the barriers to an existing research direction.

**Paper Topic And Main Contributions:**

The authors propose an induction-augmented framework that utilizes inductive knowledge derived from LLMs and the retrieved documents for better implicit reasoning. Specifically, they enhance the conventional RAG with an inductor that generates inductive knowledge.

The authors propose an IAG-GPT model which directly utilizes GPT-3 and IAG-Student which is first trained via knowledge distillation with GPT-3 pseudo labels, and then optimized through a differentiable beam search algorithm.

The experiments show that IAG-GPT has significant advantages over ChatGPT and performs extremely well on CSQA2.0 and StrategyQA. IAG-Student outperforms RAG baselines.


**Questions For The Authors:**

1. In the IAG-student algorithm, the generator is first trained followed by the inductor, will finetuning the generator together with the inductor help?

2. In line 228, How to calculate the mu and sigma for the distribution?



**Reasons To Accept:**

This paper addresses a non-trivial problem for implicit reasoning by introducing an inductor with the assistance of LLMs.

The authors conduct extensive experiments and the experiment results are reasonable.



**Reasons To Reject:**

1. The generalization ability of the model is a major concern.
- The model assigns inductive information to every question, even when some questions do not require it. A potential solution could be implementing a question classifier to identify the type of question and determine whether inductive information is necessary for a particular query.
- The strict structure/formulation of the prompt, especially the Knowledge part, is another issue (lines 245-247).

2. Another minor issue is that there is a huge gap between the performance of IAG-GPT and IAG-Student, which makes the distilled model and its corresponding algorithm less convincing. More experiments on larger models are expected.



**Reproducibility:**

4: Could mostly reproduce the results, but there may be some variation because of sample variance or minor variations in their interpretation of the protocol or method.

**Reviewer Confidence:**

4: Quite sure. I tried to check the important points carefully. It's unlikely, though conceivable, that I missed something that should affect my ratings.

---

> ### Author Rebuttal · Authors · 2023-08-26
>
> We sincerely appreciate the invaluable insights and suggestions you have offered on our paper. We have carefully reviewed these comments and tried our best to address your concerns. The point-to-point response is listed as follows.
>
> 1. **Comment**: "The generalization ability of the model is a major concern. The model assigns inductive information to every question, even when some questions do not require it. A potential solution could be implementing a question classifier to identify the type of question and determine whether inductive information is necessary for a particular query."
>
>     **Response**: Thank you for your insightful comment on the generalization of the method. You are right that not all questions require inductive information to answer. However, from our point of view, whether to use the inductive information depends not only on the question itself but also on the quality of the inductive knowledge. If we want to build a classifier to accurately determine if a question can be answered by the generator without introducing extra knowledge, does it mean that the classifier should have the same amount of knowledge as the generator? We think it might be more feasible to insert a validation module AFTER the induction module for validating the quality of the retrieved information and the inductive knowledge (e.g., detecting logically inconsistent or false statements and removing them), which we leave for our future work. However, we do agree that, from the engineering perspective, pre-detection of the necessity of induction can reduce system costs. This can be an important consideration for engineering implementations.
>
>
> 2. **Comment**: "The strict structure/formulation of the prompt, especially the Knowledge part, is another issue (lines 245-247)."
>
>     **Response**: We appreciate your comment regarding the formulation of the prompt. We admit that our formulation of the inductive prompting doesn't seem too strict. Actually, given that the input questions are free-form text, it can be hard to present the reasoning process into a well-structured formulation. Just like Chain-of-thought prompting, the reasoning path can be constructed very differently for different kinds of questions. At the core of inductive prompting is the logical reasoning that projects the target into its conceptual hypernym by analogy, which has been explicitly demonstrated in the knowledge part (i.e., "Target, analog#1, analog#2 are hypernym"). Besides, we present a few examples of inductive prompting in Table 5, which will hopefully help the readers better understand the prompting method. Nonetheless, we agree that the formalization of the research problem can benefit from a more strict formulation of the reasoning path, and we'll keep exploring a better representation of the prompting structure, both for our inductive prompting and for other methods such as CoT.
>
> 3. **Comment**: "Another minor issue is that there is a huge gap between the performance of IAG-GPT and IAG-Student, which makes the distilled model and its corresponding algorithm less convincing. More experiments on larger models are expected."
>
>     **Response**: We fully acknowledge the point that larger induction models can narrow the performance gap. It's natural to expect a huge performance gap given that our student induction model (770m) is several orders of magnitude smaller than GPT-3 (170B). But our intention of introducing IAG-Student is not to make it comparable to the IAG-GPT, but to demonstrate that using a small add-on module can improve upon the RAG baseline. Besides, we have to admit that our current GPU infrastructure doesn't support training a 13B generator and a larger induction model simultaneously. As said in the limitations section, experimenting with larger models will be the direction of our future exploration.
>
> 4. **Comment**: "In the IAG-student algorithm, the generator is first trained followed by the inductor, will finetuning the generator together with the inductor help?"
>
>     **Response**: Thanks for your insightful suggestion. In fact, we did conduct experiments that train both modules together, but it converges slower and the performance doesn't improve. In the first training stage, the generator is used to provide feedback signals for the student induction model to know which knowledge statement is better. If the generator is not finetuned in advance, it can give false feedbacks. We suppose this can be the reason for slower convergence and we decide to fine-tune the generator as the first step. I hope this explanation can address your concerns.
>
> 5. **Comment**: "In line 228, How to calculate the mu and sigma for the distribution?"
>
>     **Response**: We would like to apologize for not adequately describing the details. The distribution is a vector of scalar where each scalar represents the probability of generating the golden answer given the corresponding knowledge statement. $\mu$ and $\sigma$ are the average and deviation of this distribution respectively and can be easily calculated in the usual way: $\mu = \frac{\sum_{i=1}^{n}}{n}$ and $\sigma=\sqrt{\frac{\sum_{i=1}^n(x_i-\mu)^2}{n}}$. In light of this feedback, we'll revise this paragraph to provide a clearer explanation of the method.

---

### Official Review · Reviewer_i5uE · 2023-08-08

**Soundness:** 3

**Excitement:**

4: Strong: This paper deepens the understanding of some phenomenon or lowers the barriers to an existing research direction.

**Paper Topic And Main Contributions:**

This paper introduces the Induction-Augmented Generation framework, which integrates inductive knowledge statements for Open-Domain QA tasks to enhance implicit reasoning. The framework proposes two models: IAG-GPT and IAG-student. IAG-GPT generates inductive knowledge statements using GPT-3 and utilizes both generated knowledge prompts and retrieved documents for QA tasks. Since IAG-GPT is dependent on the GPT-3 API, IAG-student is trained to distill GPT-3.

**Questions For The Authors:**

* Is the concatenation style of Retrieval Only in IAG the same as FiD [1]? If it is FiD-style, then there will be N number of passages to encode with the query. If not, as shown in concept figure 2, there will be a single long input which is a concatenation of the query and all N passages for IAG-GPT and IAG-student. It needs more clarification because in the knowledge fusion experiment, it seems like all of the question, knowledge statements, and retrieved documents are concatenated into a single long input.
* On the CSQA dev dataset, it is stated that top-5 snippets are used (Line 423). However, in Line 465, on CSQA, it uses 10 retrieved documents. Is the top-5 snippets transformed into 10 retrieved documents? Or does it use top-10 snippets?
* Inductive prompting generates implicit knowledge through in-context learning of 5 demonstrations. How are trivial and CoT prompting done? The examples are shown in Table 7, but it is not clear how the prompts are formed.
* Are there cases where inductive knowledge statements are hallucinated?

[1] Leveraging Passage Retrieval with Generative Models for Open Domain Question Answering

**Reasons To Accept:**

The proposed approach achieved state-of-the-art performance by integrating inductive knowledge into the prompt on the CSQA and Strategy QA datasets. The presentation of quality examples (such as Table 6 and 7) further supports the validity of the work.

**Reasons To Reject:**

* The consistency in reporting the results needs to be done. Figure 3-4 and Table 2-4 uses StrategyQA dev which makes it hard to compare with Table 1 StrategyQA test baselines.
* Table 2 shows that knowledge statements generated with inductive prompting support QA performance. However, to fully verify the effectiveness of inductive knowledge statements, additional comparisons need to be made on CSQA dev with 15 retrieved documents versus CSQA dev with 10 retrieved documents and 5 inductive knowledge statements. On StrategyQA, a comparison between 10 retrieved documents and 5 retrieved documents with 5 inductive knowledge statements needs to be conducted.

**Reproducibility:**

3: Could reproduce the results with some difficulty. The settings of parameters are underspecified or subjectively determined; the training/evaluation data are not widely available.

**Reviewer Confidence:**

3: Pretty sure, but there's a chance I missed something. Although I have a good feel for this area in general, I did not carefully check the paper's details, e.g., the math, experimental design, or novelty.

**Typos Grammar Style And Presentation Improvements:**

* In Line 467, naming of the dataset should be consistent
	* sudden acronym for StrategyQA -> SQA
* Including the model size could strengthen your Table 1.

---

> ### Author Rebuttal · Authors · 2023-08-26
>
> We're grateful for the time and effort that you have taken to evaluate our work. Your expertise and attention to details have greatly contributed to the improvement of our paper. In this response, we would like to provide a detailed response to each of the raise points.
>
>
> 1. **Comment**: "The consistency in reporting the results needs to be done. Figure 3-4 and Table 2-4 uses StrategyQA dev which makes it hard to compare with Table 1 StrategyQA test baselines."
>
>     **Response**:  We sincerely apologize for any confusion or misunderstanding caused by our previous explanation/representation. Since both CSQA2.0 and StrategyQA provide blind test datasets, we have to upload our model predictions to the leaderboard website for evaluation, and re-submission is not allowed within 7 days to prevent overfitting. To conform to the policy, we decided to report our best results on the test dataset in Section 5.1 and perform the rest experiments on the development set for convenient local evaluation. But we emphasize that the scores reported for each experiment are consistent and can serve the experimental purposes.
>
> 2. **Comment**: "Table 2 shows that knowledge statements generated with inductive prompting support QA performance. However, to fully verify the effectiveness of inductive knowledge statements, additional comparisons need to be made on CSQA dev with 15 retrieved documents versus CSQA dev with 10 retrieved documents and 5 inductive knowledge statements. On StrategyQA, a comparison between 10 retrieved documents and 5 retrieved documents with 5 inductive knowledge statements needs to be conducted."
>
>     **Response**: Thanks for providing such valuable suggestions. We totally agree that it's important to compare IAG and the baseline with the same number of supporting evidence. As you point out, we conducted some extra experiments. For both datasets, we compare 10 retrieved documents and the combination of 5 documents + 5 knowledge statements. The results are as follows:
>
>     | Method | CSQA2.0 | StrategyQA |
>     |---|---|---|
>     | 10 retrievals | 78.4 | 73.8 |
>     | 5 retrievals + 5 knowledge | 78.2 | 76.2 |
>
>     As shown, for CSQA2.0, using 10 retrievals is slightly better, and for StrategyQA, using 5 retrievals and 5 knowledge statements is much better. This finding is consistent with our previous conclusion (lines 823-836) that our method works better for reasoning-intensive QA tasks. For CSQA2.0, many commonsense questions can be solved by accessing the powerful search engine such as Google. But for StrategyQA, the answers are unlikely to be directly found in documented records, and introducing inductive knowledge becomes very useful.
>
> 3. **Comment**: "Is the concatenation style of Retrieval Only in IAG the same as FiD [1]? If it is FiD-style, then there will be N number of passages to encode with the query. If not, as shown in concept figure 2, there will be a single long input which is a concatenation of the query and all N passages for IAG-GPT and IAG-student. It needs more clarification because in the knowledge fusion experiment, it seems like all of the question, knowledge statements, and retrieved documents are concatenated into a single long input."
>
>     **Response**: Yes, FiD is used for all experiments. We hypothesize that the confusion is caused by our presentation of figure 2 where we concatenate the query, knowledge statements and retrievals for simplicity. But actually, each piece of evidence is individually fed into the encoder following the FiD architecture. We'll improve the figure in the revision for clearer presentation.
>
> 4. **Comment**: "On the CSQA dev dataset, it is stated that top-5 snippets are used (Line 423). However, in Line 465, on CSQA, it uses 10 retrieved documents. Is the top-5 snippets transformed into 10 retrieved documents? Or does it use top-10 snippets?"
>
>     **Response**: We sincerely regret our oversight in the clear description of the setups. In fact, using top-5 retrieved snippets is a consistent setup for all experiments, with only one exception as you point out in Line 465 where we use 10 snippets to achieve first place on the CSQA2.0 leaderboard. Since CSQA2.0 is an open-domain QA benchmark, there's no limitation to the use of external knowledge. So we enlarged the snippet number for the competition in order to achieve the best performance.
>
> 5. **Comment**: "Inductive prompting generates implicit knowledge through in-context learning of 5 demonstrations. How are trivial and CoT prompting done? The examples are shown in Table 7, but it is not clear how the prompts are formed."
>
>     **Response**: We apologize for not making the implementation details clear enough. For CoT prompting, the original CoT paper proposed a prompt used for the StrategyQA dataset, and we use exactly the same prompt. For trivial prompting, the few-shot demonstrations are constructed by directly presenting a conclusion without reasoning. For example:
>
>     **Question**: It is safe to keep wolves as pets.
>
>     **Knowledge**: Wolves are dangerous.
>
>     As you suggested, we will explain the CoT implementation more specifically and add the trivial prompting template to the appendix in the revision.
>
>
> 6. **Comment**: "Are there cases where inductive knowledge statements are hallucinated?"
>
>     **Response**: You're right, there are such cases. Inductive reasoning can hallucinate for some strange or tricky queries. For example, GPT-3 can generate the following statement:
>
>     **Question**: Cotton candy is sometimes made out of cotton?
>
>     **Knowledge**: Cotton, wool, and silk are fabrics. Cotton candy is made out of spun sugar or fabric.
>
>     In such cases, the model can be easily misled by the query and hallucinate statements that only capture superficial reasoning patterns. We will take the advice and explicitly present these cases in the revision to provide the readers with more insights into the limitations of our method.
>
> 7. **Comment**: "Typos Grammar Style And Presentation Improvements"
>
>     **Response**: Thanks for helping us find these problems. We'll fix them in the revision.

---

### Official Review · Reviewer_bY1q · 2023-08-10

**Soundness:** 4

**Excitement:**

4: Strong: This paper deepens the understanding of some phenomenon or lowers the barriers to an existing research direction.

**Paper Topic And Main Contributions:**

The paper's key contributions are as follows:

1. The paper proposes an inductive prompting method inspired by cognitive functions of inductive reasoning. This method guides language models (LLMs), specifically GPT-3, to generate knowledge statements that establish reasoning paths.

2.  The paper introduces the Induction-Augmented Generation (IAG) framework, which enhances the traditional Retrieval-Augmented Generation (RAG) architecture.

3. The paper presents two variants of the IAG framework. IAG-GPT leverages the inductive knowledge statements sampled from GPT-3 as evidence for the generator.

**Reasons To Accept:**

This paper offers several strengths and benefits:

1.The paper introduces a novel approach, Induction-Augmented Generation (IAG), which effectively combines inductive reasoning with language generation for answering implicit reasoning questions.

2.  Implicit reasoning questions pose a significant challenge for open-domain question answering systems. By focusing on this challenge, the paper contributes to solving an important problem in the field of NLP, advancing the state of the art in understanding and generating reasoned answers.

3. The paper presents a well-defined framework (IAG) and a detailed methodology for integrating inductive knowledge into the answer generation process.

**Reasons To Reject:**

there are also some potential weaknesses:

1.  The proposed Induction-Augmented Generation (IAG) framework involves multiple components, including retrieval, induction, and generation, which might make it challenging for researchers to reproduce and implement the approach.

2.  The paper heavily relies on external language models, such as GPT-3, for generating inductive knowledge and improving performance. This reliance raises concerns about the availability, cost, and access to these models, which could limit the adoption of the proposed approach by researchers with limited resources or access.


3.  While the paper highlights successful cases where inductive knowledge enhances answer prediction, it does not thoroughly analyze or discuss cases where the approach might fail or provide incorrect answers. Understanding the limitations and potential pitfalls of the IAG framework is crucial for its safe and reliable application.

**Reproducibility:**

3: Could reproduce the results with some difficulty. The settings of parameters are underspecified or subjectively determined; the training/evaluation data are not widely available.

**Reviewer Confidence:**

5: Positive that my evaluation is correct. I read the paper very carefully and I am very familiar with related work.

---

> ### Author Rebuttal · Authors · 2023-08-26
>
> We would like to express our gratitude for the reviewer's comment highlighting the overall design of our framework and experiments. The valuable insights have prompted us to reconsider our method and to further improve the presentation. In this response, we address each of the reviewer's concerns and explain our viewpoint with additional supporting information.
>
> 1. **Comment**: "The proposed Induction-Augmented Generation (IAG) framework involves multiple components, including retrieval, induction, and generation, which might make it challenging for researchers to reproduce and implement the approach."
>
>     **Response**: We acknowledge that our implementation of the IAG framework seems a little complicated, which might be due to the training protocol. We have to clarify the IAG framework itself merely adds an induction module to the conventional RAG architecture for knowledge enhancement. When applying IAG to real-life applications, the induction module can be implemented using any off-the-shelf language models (GPT series, llama2, etc.). With the induction module frozen, the rest part of the IAG framework can be optimized exactly the same way as RAG. I hope this explanation can alleviate the concerns.
>
> 2. **Comment**: "The paper heavily relies on external language models, such as GPT-3, for generating inductive knowledge and improving performance. This reliance raises concerns about the availability, cost, and access to these models, which could limit the adoption of the proposed approach by researchers with limited resources or access."
>
>     **Response**: We sincerely appreciate the review's insightful comment on the use of GPT-3. We admit that dependencies on non-open-source services might jeopardize the integrity of academic research. Nonetheless, we noticed that, with LLMs (either open-source or non-open-source) becoming more powerful and more accessible to the public, a growing body of research has been focusing on the better utilization of these models for various research domains. Although we believe it is a trend to embrace, we do agree with the reviewer's opinion that academic research should focus on their own methods instead of relying on LLMs too heavily. As an effort to alleviate this dependency, we implemented IAG-Student in our work which makes the inference process self-contained.
>
> 3. **Comment**: "While the paper highlights successful cases where inductive knowledge enhances answer prediction, it does not thoroughly analyze or discuss cases where the approach might fail or provide incorrect answers. Understanding the limitations and potential pitfalls of the IAG framework is crucial for its safe and reliable application."
>
>     **Response**: We appreciate the reviewers' keen observation that our paper may not have analyzed the limitations in more detail. There are indeed cases where IAG can fail due to the introduction of false inductive knowledge, especially for some tricky questions. For example, GPT-3 can generate the following statements:
>
>     **Question**: Cotton candy is sometimes made out of cotton?
>
>     **Answer**: Cotton, wool, and silk are fabrics. Cotton candy is made out of spun sugar or fabric.
>
>     This false induction might be caused by a superficial imitation of the reasoning pattern and the model's inability to capture the inner logic of such a tricky question. Although our previously submitted paper did include some cases (e.g., for Q3 in Table 6, GPT-3 generates logically conflicting statements) showing the weakness of our method, we overlooked the importance of an in-depth analysis. In light of this feedback, we'll incorporate more of such cases and analysis into the revision.

---

### Official Review · Reviewer_1Ny9 · 2023-08-11

**Soundness:** 3

**Excitement:**

3: Ambivalent: It has merits (e.g., it reports state-of-the-art results, the idea is nice), but there are key weaknesses (e.g., it describes incremental work), and it can significantly benefit from another round of revision. However, I won't object to accepting it if my co-reviewers champion it.

**Paper Topic And Main Contributions:**

This paper is based on Induction Augmented Generation Framework that uses LLMs for the implicit reasoning approach. The framework outperforms baselines for Retrieval Augmented Generation and ChatGPT on two Open domain tasks.

Overall Contributions :
1) Novel inductive prompting method which improves the factuality of knowledge elicited from LLMs.
2)  A GPT implementation of the framework that improves over strong baseline models and ChatGPT.
3) A TAILBACK optimization algorithm that trains the inductor which allows IAG-Student to outperform Retrieval Augmented Generation baselines.

**Questions For The Authors:**

1) Can you explain the tailback workflow using a pseudo code with detailed steps? Currently, it's a bit difficult to follow.
2) Are there any plan to switch to later GPT versions from GPT-3? What would be the implications in terms of evaluation outcomes ?

**Reasons To Accept:**

The description of the methodology and building up the framework was well explained.
Evaluation conducted on two large Open-domain QA benchmark datasets.


**Reasons To Reject:**

1) Although the Student Inductor model is shown to surpass the benchmark, the explanation and the underlying working principle was a bit hard to follow.
2) For tailback differential beam searching is used, but it is hard to follow what are the steps pertaining to it.

**Reproducibility:**

4: Could mostly reproduce the results, but there may be some variation because of sample variance or minor variations in their interpretation of the protocol or method.

**Reviewer Confidence:**

4: Quite sure. I tried to check the important points carefully. It's unlikely, though conceivable, that I missed something that should affect my ratings.

---

> ### Author Rebuttal · Authors · 2023-08-26
>
> We would like to thank the reviewer for the insightful comments. We have carefully read these comments and realized that our paper should have presented the ideas and methods in a more clear way to alleviate these concerns. We have revised the paper by incorporating the reviewer's suggestions. Here's our point-to-point response to the reviewer's comments.
>
> 1. **Comment**: "Although the Student Inductor model is shown to surpass the benchmark, the explanation and the underlying working principle was a bit hard to follow"
>
>     **Response**: We appreciate the reviewer's observation and our paper may not have clearly described this module. In fact, the student inductor model can be regarded as a cheap substitution of GPT-3. It is implemented as a seq2seq model that takes in the question and outputs knowledge statements. The model is firstly trained by imitating GPT-3, i.e., we use the statements generated by GPT-3 to finetune the model. Next, we further optimize the model using TailBack. We will explain TailBack in more detail, which is also presented in the response to the thrid comment.
>
> 2. **Comment**: "For tailback differential beam searching is used, but it is hard to follow what are the steps pertaining to it."
>
>     **Response**: We apologize for not presenting our method as clearly as intended. We find that the beam scores reflect the probabilities of generating each token during decoding and thus can be used for constructing the loss function. Since Pytorch's default implementation of beam search is not intended for back-propagating any loss after the generation is done, the intermediate process for calculating the scores turns matrices into scalars. Differentiable beam search is in fact a modification to the Pytorch's implementation. We keep the intermediate variables in its matrix form so that the calculated beam scores are also pytorch matrices instead of scalars. As a result, the beam scores can be used for the loss function and be back-propagated through the induction model. In light of this feedback, we will add a further description of the method to the revision.
>
> 3. **Comment**: "Can you explain the tailback workflow using a pseudo code with detailed steps? Currently, it's a bit difficult to follow."
>
>     **Response**: Sure, the pseudo code is presented as follows:
>
>     \=======================================================
>
>     Procedure TailBack
>
>     $\mathcal{D}$: trainig dataset
>
>     $Rtv$: the retriever module
>
>     $\theta_{gen}$: the generator
>
>     $\theta_{ind}$: the induction model
>
>     for i in {1...num_iterations}:
>
> &emsp;&emsp; &emsp; // get a question-answer pair from the training set
>
> &emsp;&emsp; &emsp; $ (q, a*) \leftarrow getSample(\mathcal{D}) $
>
> &emsp;&emsp; &emsp; // induction module generates N statements based on the question, also get the beam scores of each statement.
>
> &emsp;&emsp; &emsp; $(K_1, s_1), (K_2, s_2), ..., (K_N, s_N) \leftarrow diffBeamSearch(\theta_{ind}, q) $
>
> &emsp;&emsp; &emsp; // get M documents from the retriever
>
> &emsp;&emsp; &emsp; $\{R_1, R_2, ..., R_M\} \leftarrow Rtv(q)$
>
> &emsp;&emsp; &emsp; // use generator to calculate the possibilities of generating the golden answer
>
> &emsp;&emsp; &emsp;for K_i in {K_1, K2, ..., K_N}:
>
> &emsp;&emsp; &emsp; &emsp;&emsp;$p_i \leftarrow seq2seq\_loss(\theta_{gen}, q, K_i, {R_1, R_2, ..., R_M}, a*)$
>
> &emsp;&emsp; &emsp;end
>
> &emsp;&emsp; &emsp;// normalize the possibilities into confidence values using equation(2)
>
> &emsp;&emsp; &emsp;$\{\tilde{p_1}, \tilde{p_2}, ..., \tilde{p_N}\} = normalize({p_1, p_2, ..., p_N})$
>
> &emsp;&emsp; &emsp;// calculator the loss value for the induction model
>
> &emsp;&emsp; &emsp;$loss = \sum_{i=1}^N \tilde{p_i} * s_i$
>
> &emsp;&emsp; &emsp;$loss.backward()$
>
> &emsp;end
>
> &emsp;\=======================================================
>
> &emsp;As you suggest, we will add the pseudo code into the revision if the space permits.
>
>
> 4. **Comment**: "Are there any plan to switch to later GPT versions from GPT-3? What would be the implications in terms of evaluation outcomes ?"
>
>     **Response**: We fully acknowledge the point that experimenting with more LLMs will attach more valuable insights to our paper. In fact, after OpenAI made the API for GPT-3.5 publicly available, we conducted a few extra experiments on the development set. However, we didn't observe consistent performance improvement. The reason might be that GPT-3.5 is better at following human instructions, but its reasoning ability doesn't improve that much. As for GPT-4, we noticed that it has achieved first place in some nlp leaderboards. Although we currently don't have access to GPT-4, we expect the knowledge statements generated by GPT-4 to be much more informative and helpful.

---

### Meta-Review · Area_Chair_nLPG · 2023-09-19

**Recommendation:** 4

**Metareview:**

This paper proposes the Induction-Augmented Generation (IAG) framework, designed to enhance implicit reasoning in Open-Domain Question-Answering (QA) tasks. The framework includes two models: IAG-GPT and IAG-Student. IAG-GPT generates inductive knowledge statements using GPT-3 and combines these statements with retrieved documents for QA tasks. Removing the dependency on GPT-3 at inference time, the IAG-Student model is trained through knowledge distillation with GPT-3 pseudo labels and optimized using a differentiable beam search algorithm.

This paper presents a well-explained methodology and framework for solving the challenging problem of implicit reasoning in open-domain question answering. This work proposes a novel optimization scheme to train the inductor model through distillation and back-propagation of the generator feedback via differentiable beam scores. The authors conduct a convincing evaluation on two open-domain QA benchmark datasets (CSQA2 and Strategy QA) and achieve state-of-the-art performance on both (at least as of January 2023 for Strategy QA).

As highlighted by the reviewers, the authors could further strengthen their contribution by conducting, for example, a more thorough analysis of the failure modes of the approach, of the generalization of the method, and of the effectiveness of the inductive knowledge statements.

---

### Decision · Program_Chairs · 2023-10-07

**Decision:**

Accept-Main

**Comment:**

This paper proposes the Induction-Augmented Generation (IAG) framework, designed to enhance implicit reasoning in Open-Domain Question-Answering (QA) tasks. The framework includes two models: IAG-GPT and IAG-Student. IAG-GPT generates inductive knowledge statements using GPT-3 and combines these statements with retrieved documents for QA tasks. Removing the dependency on GPT-3 at inference time, the IAG-Student model is trained through knowledge distillation with GPT-3 pseudo labels and optimized using a differentiable beam search algorithm.

This paper presents a well-explained methodology and framework for solving the challenging problem of implicit reasoning in open-domain question answering. This work proposes a novel optimization scheme to train the inductor model through distillation and back-propagation of the generator feedback via differentiable beam scores. The authors conduct a convincing evaluation on two open-domain QA benchmark datasets (CSQA2 and Strategy QA) and achieve state-of-the-art performance on both (at least as of January 2023 for Strategy QA).

As highlighted by the reviewers, the authors could further strengthen their contribution by conducting, for example, a more thorough analysis of the failure modes of the approach, of the generalization of the method, and of the effectiveness of the inductive knowledge statements.